# Energy-ordered resource stratification as an agnostic signature of life

Akshit Goyal [1] ✉ & Mikhail Tikhonov [2] ✉

The search for extraterrestrial life hinges on identifying biosignatures, often focusing on gaseous metabolic byproducts as indicators. However, most such biosignatures require assuming specific metabolic processes. It is widely recognized that life on other planets may not resemble that of Earth, but identifying biosignatures "agnostic" to such assumptions has remained a challenge. Here, we propose a novel approach by considering the generic outcome of life: the formation of competing ecosystems. We use a minimal model to argue that the presence of ecosystem-level dynamics, characterized by ecological interactions and resource competition, may yield biosignatures independent of specific metabolic activities. Specifically, we propose the emergent stratification of chemical resources in order of decreasing energy content as a candidate new biosignature. While likely inaccessible to remote sensing, this signature could be relevant for sample return missions, or for detection of ancient signatures of life on Earth itself.

The search for extraterrestrial life hinges on the identification of biosignatures, markers indicating the presence of biotic processes[1]. By necessity, all proposed candidates are inspired by the kind of life we know − the forms of life as they exist on Earth. Perhaps the most commonly discussed biosignature is the expected presence of certain gaseous metabolic byproducts[2,3], which could be detected spectroscopically, with much attention devoted to specifically molecular oxygen[4]. This kind of signatures assumes that the biotic processes we seek to detect employ specific chemical transformations for their metabolism.

It is widely recognized that life elsewhere in the universe need not resemble the terrestrial form, motivating the interest in so-called agnostic biosignatures, those that are "not tied to a particular metabolism-informational biopolymer or other characteristic of life as we know it"[5–7]. Some proposals include looking for polyelectrolytes[7] or homochirality[8], but identifying agnostic biosignatures is a serious challenge. In fact, it has been proposed that truly agnostic signatures may not even exist[9]. All proposed signatures require some additional assumptions, e.g., metabolism, morphology, chirality. Even setting aside the technological challenges of remote sensing, and assuming we could perform arbitrary measurements (e.g., to detect ancient signatures of life on Earth, which is also an active field of research), it is

not clear what to look for without making such assumptions. While we all agree that life requires self-replication supporting a Darwinian process, no measurable signature is known to be a generic consequence of self-replication alone.

One feature widely recognized as a universal attribute of life is the consumption and transformation of energy. This observation is at the heart of much astrobiology-related experimentation, such as studies of low-energy life, and the diversity of energy harvesting mechanisms. It was previously suggested that energy-based considerations could also help identify new biosignatures, based on the existence and maintenance of chemical imbalances, or the direction and magnitudes of energy fluxes (the "follow the energy" approach[10]). However, this requires a criterion distinguishing the outcomes of biotic processes from what is achievable abiotically. It remained unclear whether such a distinguishing criterion could exist independently of the details of a specific energy harvesting mechanism; in other words, whether such a signature could ever be truly agnostic. To our knowledge, no such criterion has been proposed.

In this work, we combine the energy perspective with the observation that, as far as we know, life forms never exist alone, but generically develop into systems characterized by ecological interactions and resource competition (even in experimental conditions

[1]International Centre for Theoretical Sciences, Tata Institute of Fundamental Research, Bengaluru, India. [2]Department of Physics, Washington University in St Louis, St. Louis, MO, USA. ✉e-mail: akshitg@icts.res.in; tikhonov@wustl.edu

specifically intended to avoid this[11]). On Earth, there is only one known exception[12], and even in this example, the genome of the lone organism shows clear evidence of horizontal gene transfer, indicating it was part of an ecology in its past. This suggests that the assumption of life forming an ecosystem is almost as general as the minimal requirement, that of a Darwinian process.

The relevance of the ecological perspective for astrobiological research has been recognized, e.g., in the studies of elemental ratios, serpentinization, or metabiospheres[13–17]. Here, we propose that this generic emergence of competing ecologies can lead to an agnostic energetic biosignature: specifically, the emergent spatial stratification of chemical resources in order of their energy content (Fig. 1). (To avoid confusion, we would like to contrast this with the term "energy gradient" in the astrobiological literature. That term refers to the fact that sustaining life requires the presence of a compound at a chemical disequilibrium, which could be harnessed for energy. This "gradient" is in energy space. In contrast, in this work we are talking about a *spatial* gradient of multiple energy-containing compounds.) Such patterns are observed in many contexts on Earth[18–23]. Abiotically, there is no reason to expect this, as the abiotic rate of a chemical reaction and its energy yield are set by independent parameters (the height of the activation barrier and the energy of the final state, respectively), and are generically uncorrelated. Using a minimal theoretical model, we demonstrate that energy-ordered stratification is a robust consequence of two processes: biological self-replication as species consume resources, and ecological interactions between different biological species as they compete for space. Our model does not assume any specific molecular detail or metabolism. Thus, we propose energy-ordered resource stratification as a candidate agnostic biosignature requiring minimal assumptions on the chemical implementation of the Darwinian process.

## Results

### The model

We seek to understand the consequences of self-replication and ecological interactions for biosignatures. To this end, we consider the following minimal model implementing these two ingredients (Fig. 2a) in a simple setting. We track the dynamics of the abundance of $S$ reaction catalysts $N_i(x, t)$ and $M$ chemical resources $R_\alpha(x, t)$ along a one-dimensional spatial coordinate $x$ (representing, e.g., depth in a microbial mat or water column). All reaction catalysts consume different resources and diffuse over space with diffusion constant $D_N$. A global parameter $\gamma$ controls whether reaction catalysts are biotic and can self-replicate: $\gamma = 0$ corresponds to abiotic catalysis, while $\gamma \neq 0$ corresponds to self-replicating biological species, which grow as they consume resources. To maintain the growth of these species,

resources are supplied abiotically from the outside at $x = 0$ at a constant flux $K_\alpha$. Thus, over time, resources are supplied, depleted and diffuse over space with diffusion constant $D_R$. For a simple implementation of ecology, we assume that biological species at the same location $x$ also compete with each other with a pairwise competitive interaction strength $A_{ij}N_i(x, t)N_j(x, t)$. This competition could arise due to e.g. physical contact-based inhibitory interactions[24] or competition for space[25,26]. A global parameter $\rho$ controls the density of ecological interactions by controlling the fraction of non-zero entries in the matrix $A_{ij}$; $\rho = 0$ corresponds to the case of no ecology. In this manuscript, we will only vary the key parameters $\gamma$ and $\rho$, representing self-replication and ecology, while keeping the rest fixed. Finally, we assume for simplicity that neither species nor resources can leave the system. With these assumptions, the equations governing the dynamics of our model can be expressed as follows:

$$\frac{\partial N_i(x, t)}{\partial t} = N_i(x, t)\left(\gamma g_i(\vec{R}) - \sum_{j \neq i} A_{ij}N_j(x, t)\right) + D_N \nabla^2 N_i(x, t) \tag{1}$$

$$\frac{\partial R_\alpha(x, t)}{\partial t} = -\sum_i f_{i\alpha}(\vec{N}, \vec{R}) + D_R \nabla^2 R_\alpha(x, t) \tag{2}$$

$$-\nabla R_\alpha(0, t) = K_\alpha - \sum_i f_{i\alpha}(\vec{N}(0, t), \vec{R}(0, t)) \tag{3}$$

$$-\nabla R_\alpha(L, t) = -\sum_i f_{i\alpha}(\vec{N}(L, t), \vec{R}(L, t)) \tag{4}$$

Here, the self-replication parameter $\gamma$ controls the degree of self-replication ($\gamma = 0$ meaning no self-replication), and the growth of biological species $g_i$ and consumption of resources $f_{i\alpha}$ have the following functional forms:

$$g_i(\vec{R}) = \sum_{\alpha=1}^{M} Y_\alpha k_{i\alpha} R_\alpha(x, t) - m_i, \tag{5}$$

$$f_{i\alpha}(\vec{N}, \vec{R}) = k_{i\alpha} R_\alpha(x, t) N_i(x, t). \tag{6}$$

The parameter $m_i$ represents the maintenance energy for species $i$, $Y_\alpha$ represents the energy yield obtained by utilizing a unit of resource $\alpha$, and $k_{i\alpha}$ is the matrix describing the resource consumption rates of each species. For simplicity of presentation, we will first assume that each biological species is a specialist and consumes only one resource unique to each species ($M = S$, and $k_{i\alpha} = \delta_{i\alpha}$). Our results are robust to relaxing these assumptions, as will be discussed later.

The foundational assumption that we will use throughout the discussion below is that the "usable energy" an organism can extract from a compound (the energy yield $Y_\alpha$) correlates with the total internal energy of this compound. There are two reasons why these two quantities are, in general, distinct: First, whether a reaction is thermodynamically favorable depends not only on energy, but also on the concentrations (of both reactants and products). Second, organisms may not be able to utilize the internal energy of all resources equally efficiently; e.g., on Earth, it took evolution millions of years to "learn" to utilize lignin, abundant during the Carboniferous period. Nevertheless, higher-energy compounds are at least in principle capable of sustaining higher yields, and one expects this correlation to increase during the course of metabolic evolution.

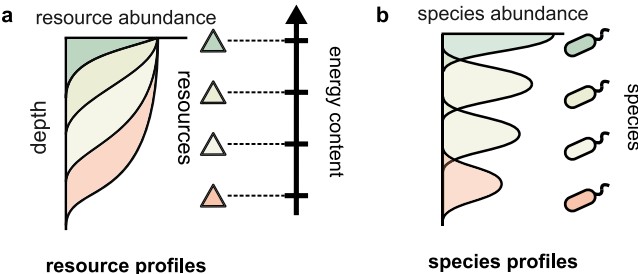

**Fig. 1 | Energy-ordered resource stratification as an observable signature of life. a** Stratified profiles of chemical resources layered by energy content are commonly observed on Earth, e.g., microbial mats, early Earth fossils (stromatolites), Winogradsky columns, and in marine environments. **b** Such profiles are generally understood to be shaped by biotic species (typically microbes) that metabolize these resources for energy. Here, we propose that energy-ordered resource stratification is a robust signature of biotic action.

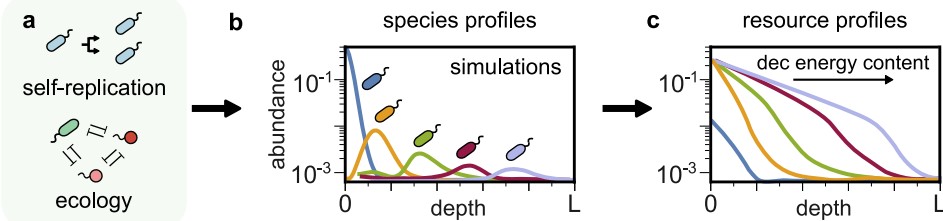

**Fig. 2 | Self-replication and ecology lead to energy-ordered resource stratification. a** Two universal features of life are self-replication and ecological interactions between different biological species—the simplest being antagonism. **b** Simulating a minimal model incorporating these two ingredients (for details see text) shows that these two ingredients lead to spatially stratified profiles of (**b**) species and (**c**) resources. Shown here is an example from a simulation for 5 species and 5 resources. Antagonistic interactions segregate species spatially, with species displacement order determined by the energy content of the resource they consume. In each segregated zone, species deplete resources proportional to their abundance.

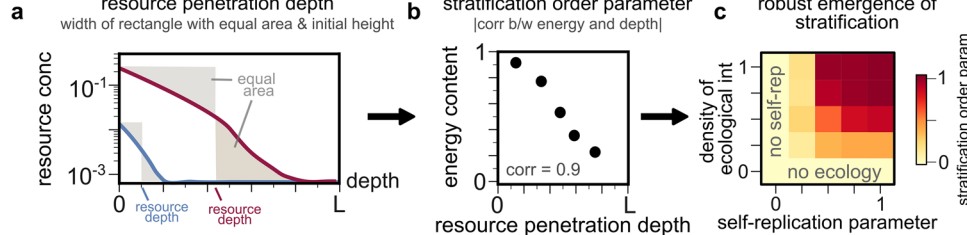

**Fig. 3 | Both self-replication and interspecies antagonism are necessary for robust spatial stratification. a** Quantification of resource penetration depth: for each simulated resource profile (blue and red), we find the width of the rectangle with area equal to that of the resource profile and the same initial height. **b** Quantification of stratification order parameter: for all resource profiles obtained from one simulation, we compute the negative of the correlation between their penetration depths and energy content $Y_\alpha$ (shown is an example from a simulation with 5 profiles). **c** Heatmap of the stratification order parameter over multiple simulations, where we systematically varied the self-replication parameter $\gamma$ and the density of ecological interactions $\rho$. Spatial stratification does not emerge in the absence of either self-replication or ecology. As $\gamma$ and $\rho$ both increase, stratification emerges robustly.

## Emergence of energy-ordered resource stratification

Simulations of our model show that self-replication and ecology lead to spatial stratification of biological species and chemical resources (Fig. 2b, c). Starting from no spatial stratification, with homogeneous species profiles and identical resource profiles, species and resource dynamics naturally converge towards a steady state where the biological species, and consequently resources, become spatially stratified (Fig. 2b, c). The emergence of this stratification can be understood as follows: the species using the most energetic resource grows the fastest near the source $x = 0$. Because it grows fastest, this species most strongly antagonizes all others near $x = 0$ (Fig. 2b, blue). As a result, less energetic resources are not consumed near $x = 0$ and penetrate further (Fig. 2b, orange, green, red and purple). The species using the next most energetic resource (Fig. 2b, orange) then grows the fastest in the adjacent region, and similarly inhibits the growth of others (Fig. 2b, green). This process continues as inhibited species and unconsumed resources diffuse further away from the source.

The resulting pattern of resource profiles is similarly spatially stratified, as resources with progressively decreasing energy content are depleted deeper and deeper away from the source $x = 0$ (Fig. 2c). We refer to this spatial pattern as energy-ordered resource stratification.

To quantify the degree of energy-ordered spatial stratification, we first compute the penetration depth of each resource, defined as the width of the rectangle with the same height as the resource concentration at the source $x = 0$, and with an area equal to that of the resource profile (quantified numerically; pictorial representation in Fig. 3a). At the penetration depth, the area of the rectangle which does not overlap with the resource profile has the same area as the remaining resource profile (Fig. 3a; shaded area). For each simulation, we measure the "stratification order parameter" as the (negative of) the correlation between the energy content and penetration depth of all simulated resources (so that stratification in order of *decreasing* energy content corresponds to a positive order parameter; Fig. 3b).

To test the robustness of energy-ordered spatial stratification, we repeat 1000 simulations of our model across randomly chosen conditions (see Methods). Throughout simulations, we systematically vary two key parameters: the self-replication parameter $\gamma$ and the density of ecological interactions $\rho$. In each case, we quantify the mean stratification order parameter across simulations.

We find that self-replication and ecology are not only sufficient, but also necessary to generate energy-ordered stratification (Fig. 3c): In the absence of self-replication ($\gamma = 0$), the energy content of a resource has no bearing on its spatial profile, with penetration depth set by diffusion and consumption rate (not energy content). In the absence of ecology ($\rho = 0$), species coexist with no spatial segregation; as a result, all resources are co-utilized to depletion, with energy-ordered stratification again failing to emerge. In the presence of both, energy-ordered stratification emerges robustly, with the stratification order parameter rapidly transitioning to $\approx 1$ as both $\gamma$ and $\rho$ increase (Fig. 3c; dark red region). Taken together, self-replication and ecology are both sufficient and necessary to generate self-organized spatial stratification of resources in order of their usable energy content.

In the interest of clarity, the model discussed above made multiple simplifying assumptions. In particular, we assumed that each species consumes only one resource (no generalism), each resource is consumed by only one species (no redundancy), and that all resources are supplied externally (no cross-feeding). The supplementary sections confirm that energy-ordered resource stratification is robust to relaxing all three assumptions: allowing cross-feeding, where only one resource is externally supplied and others are generated through metabolic byproducts (Fig. S1 and Supplementary Note 1); allowing multiple species to consume the same resource (Fig. S2 and

Supplementary Note 2); and allowing species to be generalists (Fig. S3 and Supplementary Note 3).

## Discussion

In the search for extraterrestrial life, the challenges are not only technological, but also conceptual. What measurable signatures can be expected from life *just* because it's life? Even if "measurable" is interpreted (as we do here) as "measurable in principle," not "measurable remotely" or "measurable with current technological capabilities," this question—identifying agnostic biosignatures—remains very difficult.

Life universally requires energy, and it has been noted that expenditures of energy could lead to "physically or chemically ordered systems or structures" which could serve as biosignatures[10,27,28]. However, abiotic processes can also lead to intricately ordered structures[29,30], and no mechanism-independent criterion capable of distinguishing biotic and abiotically generated order has been proposed. Here, we combine the energy perspective with an ecological perspective to propose one such candidate criterion: spatial stratification of chemical compounds in order of their intrinsic energy content.

We showed that the combined action of two ingredients widely believed to be universal attributes of life—self-replication and interspecies antagonism—naturally leads to such a self-organized stratification. This is because (1) self-replication produces an emergent correlation between the energetic yield of a resource and its rate of depletion; and (2) the interspecies antagonism then allows metabolic processes to become spatially segregated. We used a minimal model to argue that these ingredients are both sufficient and necessary. This mechanism does not depend on specific hypotheses about the chemical implementation of the organisms or their metabolism, but only on the plausible assumption that for a sufficiently evolved life form, the usable chemical energy in a compound correlates with its total internal energy; and on the assumption that resource supply is spatially inhomogeneous (establishing the origin from which the spatial stratification would establish itself). Thus, we propose that energy-ordered resource stratification might serve as a robust agnostic biosignature.

We note that even biosignatures associated with specific metabolisms are generally assumed to be part of ecosystem-level processes. For instance, isotopic fractionation of sulfur, discussed as a candidate biosignature[31], effectively requires an ecosystem-level sulfur cycle. In this sense, many signatures of life are already understood to be signatures of ecosystems[28,32,33]. However, the biosignatures usually considered are imprinted by some specific metabolic process this ecosystem is assumed to run. In contrast, the pattern discussed here is expected to arise from the competitive nature of ecosystem dynamics, rather than specific metabolic activities.

Stratified structures can also emerge abiotically, e.g., sedimentation and calcification[34]. However, these abiotic mechanisms are driven by solubility or other chemical properties, and are not expected to be correlated with the chemical energy in a compound. In contrast, biotically, such a correlation emerges naturally, as our model illustrates.

The key limitation of our analysis is that our conceptual focus leaves aside the question of technological feasibility. It is intriguing to speculate whether spatial resource stratification could manifest itself as a layering in a planet's atmosphere. However, beyond this limited context, the signature discussed here is likely inaccessible to remote sensing. The advent of missions allowing direct probing of Mars soil or the sample return missions such as Hayabusa or OSIRIS-REx has opened new possibilities. However, it is important to note that the current sampling methods preserve only a limited amount of spatial structure. Whether this idea can be adapted to a form compatible with the current technologies, and if so, in what context (ancient life on Earth, Solar system missions, remote sensing of exoplanets), remains an open question.

## Methods

We simulated our model by numerically evolving equations (1)–(2) with the assumptions as in equations (3) and (4). All simulations were done on a 1D domain of length $x \in [0, L]$ where $L = 100$ in arbitrary units, assuming no boundary flux Neumann boundary conditions for species, and assuming resources entered at $x = 0$ at flux $K$ and had no flux at $x = L$. For all simulations, we set $D_N = 10$, $D_R = 20$, $m_i = 0.1$ and $k_{i\alpha} = \delta_{i\alpha}$ for all species and resources. For each simulation with $M$ resources and $S = M$ species, we chose the energy content $Y_\alpha$ of each resource $\alpha$ randomly from a uniform distribution between 0 and 2. For competitive interaction strengths $A_{ij}$ between species $i$ and $j \neq i$, we first picked a fraction $\rho$ of the interactions randomly, setting the rest to zero. For the picked interaction strengths, we chose them by randomly selecting a number from a normal distribution with mean 0.4 and standard deviation 0.1. All diagonal entries $A_{ii}$ were left out of this procedure and set to zero. For initial conditions, we always set homogeneous initial conditions for species, while choosing quadratically decaying profiles for resources, numerically obtained to satisfy the boundary conditions.

## Data availability

No datasets were generated or analysed during the current study. The simulation data are reproducible with the code provided (see Code Availability statement).

## Code availability

Simulations were performed in Wolfram Mathematica 13 and the scripts sufficient to reproduce our results are publicly available[35].

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

## Acknowledgements
We are grateful to A. Murugan, R. Braakman, P. Byrne, G. Fournier and S. Seager for helpful discussions. This work was supported in part by the NSF grants PHY-2310746 and PHY-2340791. A.G. acknowledges support from the Ashok and Gita Vaish Junior Researcher Award, as well as the Government of India's DST-SERB Ramanujan Fellowship.

## Author contributions
M.T. conceived the study. A.G. performed the research. Both authors developed the methodology and wrote the manuscript.

## Competing interests
The authors declare no competing interests.
