## [Transparent Peer Review file · Nature Communications]

Energy-ordered resource stratification as an agnostic signature of life

Corresponding Author: Dr Mikhail Tikhonov

Version 0:

Reviewer comments:

Reviewer #1

(Remarks to the Author)

This paper proposes a way to detect extraterrestrial life using a biosignature that is independent of chemistry, namely the tendency of life to form ecosystems. In particular, the authors perform some suggestive numerical calculations indicating that self-replication and competition in a spatially-extended setting generically lead to spatial stratification of environmental resources, with an ordering based on the energy content of the resources. This energy-content ordered spatial stratification is an emergent pattern formation process that is different from spatial stratification arising from physical processes, because, it is hypothesized, only biotic processes will be dependent on energy content.

I think that this is an original and interesting contribution to the field of astrobiology, and recommend publication after some comments have been attended to. The main comments are about the robustness of their main result, and I recognize that subsequent studies may be required to address these comments definitively. Still, it might be nice for the authors to comments on the issues I'm raising, if they think they have merit. So these comments are optional.

(1) In real ecosystems, each species can consume more than one resource, often with a diauxic shift, as the authors know well. Does this affect their conclusions?

(2) With diauxic shift, the interaction between the species would be more complex. Does this affect the conclusions?

(3) A related question is that each environmental niche, which can be thought of as identified by nutrient source, is usually occupied by multiple species, not one. There are many proposed resolutions to this "paradox of the plankton". Does this affect their conclusions?

(4) Does the multidimensional nature of niche space affect their conclusions?

(5) The authors stratification relies on there being a one-dimensional ordering. So it might be appropriate in an environmental gradient, but I wonder if it can be extended to higher dimensions where there is no one-dimensional ordering possible? If there is an environmental gradient, how does one separate physical stratification from biotic stratification? I presume that this will be governed by the species interactions somehow. In the simulation, a gradient is imposed implicitly by the boundary conditions. So some discussion of these points might be helpful.

There are some minor suggestions for the readability of the paper.

(A) In Eq. (1), the competitive interaction between species is described by the matrix element α_{ij} . However, the resource is labeled by index α , and the use of the same symbol can cause confusion.

(B) The resource flux is written as flux K but never used in the equations directly.

(C) The consumption of resource by each species is described by the matrix element $k_{i\alpha}$, which is never explicitly explained in words, and the readers are expected to deduce its meaning. It would be clearer to state its definition.

(D) The authors talk about stratification, but I think it should be stated explicitly as "spatial stratification" to avoid any chance

of a reader thinking about other stratifications related to internal variables such as metabolism.

(E) On lines 42-45 the authors state that “assumption of life forming an ecosystem is almost as weak as the minimal required assumption of a Darwinian process.” I had to read this several times before I understood that what they really meant was that it is generically the case that life forms ecosystems. When I first read it, I thought they meant that the assumption is weak, meaning that it does not have much explanatory power. I think they should just say that they assume that life forms ecosystems, since all examples on Earth have this characteristic, even in extreme environments such as acid-mine drainage.

(F) They give the example of an exception to the notion that ecosystems exhibit species richness, which is the work of Chivian et al (2008). However, even that supposedly single species ecosystem has organisms whose genome has clear evidence of horizontal gene transfer from Archaea to Bacteria. This indicates that the organisms detected had at some point in their past, if not in the present, been part of an ecosystem. Also, I wondered if one needs to distinguish between stable and unstable ecosystems. An ecosystem on the verge of extinction might be passing through a bottleneck where there is only a single species left.

Reviewer #2

(Remarks to the Author)

In this manuscript, the authors propose a new candidate for agnostic biosignature for the search for extraterrestrial life. The accumulation of specific materials has previously been proposed as a biosignature. However, these are based on the specific metabolic pathways of life found on Earth, and whether they can be applied to extraterrestrial organisms, whose metabolic pathways may be completely different from those on Earth, is unknown. Therefore, the authors proposed a new biosignature based on microbial ecosystems, the energy-ordered resource stratification. The authors constructed a minimal model of a microbial ecosystem and clarified the conditions under which spatial resource stratification emerges.

The authors' attempt to explore biosignatures beyond material details is very important. Also, the concept of biosignatures based on ecosystems is novel. I agree with the argument that the distribution of compounds should be used as a biosignature rather than the presence of specific ones. However, the effectiveness of resource stratification as a biosignature is somewhat questionable. The authors need to address several issues.

Major:

1. First, to observe the resource stratification proposed by the authors as a biosignature, samples must be returned while maintaining a solid spatial structure, but is this even possible? The current method of sample return missions does not preserve the spatial structure. If the technology to return samples while preserving structure is established in the future, it may be possible to observe extraterrestrial organisms directly. In that case, in what situations do the authors expect the proposed biosignature to be effective?

2. The authors assumed that the growth yield on a resource correlates with its chemical energy, but how true is this assumption? Is total chemical energy the internal energy of a compound or its chemical potential? If it is the former, then even if the internal energy is high and cannot be consumed by living organisms, it will not contribute to the growth rate. If it is the latter, then the chemical potential is determined by the amount of the compound present, but since this amount itself depends on the metabolism of the organisms. Is it possible to determine the chemical potential of a compound from a sample with unknown extraterrestrial organisms?

3. In their model, the authors assume the existence of competition that does not depend on competition for nutrients (depending on $\alpha_{i,j}$). They also show that if this competition is not strong enough, i.e., without global competition between organisms, resource stratification will not appear. Various competition mechanisms that work between specific species are known, but what kind of global competition mechanism is assumed?

4. In the model, the authors assume that all species are specialists, i.e., they consume only one compound. If many species are generalists, how well will resource stratification be maintained?

Minor:

Line 118: The authors stated, "If the system were well-mixed, the single "fittest" biological species using the most energy-dense resource would outcompete all others and take over the system." However, will this happen unless p is close to 1? For example, if p is 0, the same number of species as the number of nutrients would remain.

Reviewer #3

(Remarks to the Author)

This paper is part of a series of very interesting and important recent papers on microbial ecosystem self-organization, stability, energy extraction, and environmental perturbations by Goyal, Tikhonov, et al. This body of work introduces a variety of new general theoretical perspectives to the field, and is both useful for interpreting modern ecology and evolution and for understanding important questions about the origins of life.

However, the current paper is more of an initial hypothesis suited for a more specialized journal, and I don't think meets the level of Nature Communications.

I have two major comments that should be addressed in anycase:

1) The work needs to be much better situated in the astrobiological literature:
- A classic focus in astrobiology-related experimentation and field work has been metabolic stratification in biofilms by Tori Hoehler et al. and many other types of ecosystems including ocean sediments (Bo Jorgensen, Jan Amend, Victoria Orphan, etc.). Funding studies on low-energy life, the diversity of energy harvesting mechanisms, and energetic ecologies is a long-running focus of NASA and there is much more work to cite in this study.
- There are many recent efforts focused on energy gradients as an agnostic biosignature (see the work of Peter Girguis and Jeffrey Marlow) which is the main proposal of this paper.
- Ecological perspectives on astrobiology have recently had a few important advances both conceptually and specifically that should also be interfaced with in this study. Recent prominent examples include:

a) Affholder, A., Guyot, F., Sauterey, B., Ferrière, R., & Mazevet, S. (2021). Bayesian analysis of Enceladus's plume data to assess methanogenesis. *Nature Astronomy*, 5(8), 805-814.

b) Kempes, C. P., Follows, M. J., Smith, H., Graham, H., House, C. H., & Levin, S. A. (2021). Generalized stoichiometry and biogeochemistry for astrobiological applications. *Bulletin of Mathematical Biology*, 83(7), 73.

c) Meurer, J. C., Haqq-Misra, J., & de Souza Mendonça, M. (2024). Astroecology: bridging the gap between ecology and astrobiology. *International Journal of Astrobiology*, 23, e3.

d) Cockell, C. S., Simons, M., Castillo-Rogez, J., Higgins, P. M., Kaltenecker, L., Keane, J. T., ... & Vance, S. D. (2024). Sustained and comparative habitability beyond Earth. *Nature Astronomy*, 8(1), 30-38.

2) It is not clear to me how these models are differentiated from any number of ecological models, even those used in modeling detailed global ecology. A broader discussion and explanation is needed here.

Reviewer #4

(Remarks to the Author)

Version 1:

Reviewer comments:

Reviewer #1

(Remarks to the Author)

I am satisfied with the authors' response and modifications to the manuscript. I recommend publication.

Reviewer #3

(Remarks to the Author)

The authors have done quite a lot of work in situating the current manuscript in the previous literature and responding to my previous critiques. I want to reiterate that I am a fan of the series of papers from these authors. However, I must say that I still think that this paper is more of an initial hypothesis and would be better suited for a journal like *Astrobiology*. This was not raised by the other two reviewers and I leave it to the editor to make this call.

Reviewer #4

(Remarks to the Author)

[Please note that for reviewer's convenience, all line numbers refer to the PDF highlighting the changes made during revisions, not the new manuscript itself.]

REVIEWER COMMENTS

Reviewer #1 (Remarks to the Author):

This paper proposes a way to detect extraterrestrial life using a biosignature that is independent of chemistry, namely the tendency of life to form ecosystems. In particular, the authors perform some suggestive numerical calculations indicating that self-replication and competition in a spatially-extended setting generically lead to spatial stratification of environmental resources, with an ordering based on the energy content of the resources. This energy-content ordered spatial stratification is an emergent pattern formation process that is different from spatial stratification arising from physical processes, because, it is hypothesized, only biotic processes will be dependent on energy content.

I think that this is an original and interesting contribution to the field of astrobiology, and recommend publication after some comments have been attended to. The main comments are about the robustness of their main result, and I recognize that subsequent studies may be required to address these comments definitively. Still, it might be nice for the authors to comment on the issues I'm raising, if they think they have merit. So these comments are optional.

We thank the reviewer for such a positive assessment. The question of robustness is an important one, and we leveraged these excellent suggestions to strengthen our result.

(1) In real ecosystems, each species can consume more than one resource, often with a diauxic shift, as the authors know well. Does this affect their conclusions?

This is an excellent point. We have expanded our model to allow for generalists, and show that stratification is robust for a range of the "extent of generalism parameter", interpolating between pure specialists and complete generalists (line 267, new fig. S3, lines 596-610).

(2) With diauxic shift, the interaction between the species would be more complex. Does this affect the conclusions?

This is a point we spent a long time with. In the end, we decided we couldn't address it substantively within the scope of this work. Diauxie manifests itself in a dynamical regime where species switch their activity over time (e.g. in a serial dilution). This is incompatible with the steady-state framework we adopted here for simplicity. Indeed, in a steady-state setup (such as a chemostat) many species that are typically diauxic lose their switching behavior; instead they co-utilize resources (e.g., see Okano et al, *Curr Opin in Microbiology* (2021)). Including a discussion of diauxie would require introducing an entire second modeling framework (where resources would arrive in batches, prompting boom/bust periods) which would be quite heavy and distract from the intent of the work (to propose and illustrate one simple idea).

(3) A related question is that each environmental niche, which can be thought of as identified by nutrient source, is usually occupied by multiple species, not one. There are many proposed resolutions to this “paradox of the plankton”. Does this affect their conclusions?

(4) Does the multidimensional nature of niche space affect their conclusions?

We interpreted the suggestions (3) and (4) as being closely related, both referring to the functional redundancy known to be ubiquitous in Earth ecosystems: each reaction can be run by a multitude of species, not just one (as in our original model). This is indeed an important feature to include, and we added an analysis demonstrating that our result is robust to including such redundancy into the model (lines 265-266, new Fig. S2, lines 567-595).

(5) The authors’ stratification relies on there being a one-dimensional ordering. So it might be appropriate in an environmental gradient, but I wonder if it can be extended to higher dimensions where there is no one-dimensional ordering possible? If there is an environmental gradient, how does one separate physical stratification from biotic stratification? I presume that this will be governed by the species interactions somehow. In the simulation, a gradient is imposed implicitly by the boundary conditions. So some discussion of these points might be helpful.

This is a very important point; thank you for prompting us to clarify: We do not see energy-ordered stratification as one example of a possible family of stratification-based signatures in higher dimensions. The energy axis is special [Hoehler et al 2007, new Ref 10], and is at the heart of our proposal. The fact that living objects or processes can generate highly spatially structured patterns is well-known, but the key question is indeed distinguishing them from abiotically generated patterns, which can be highly intricate. A priori, it is not obvious that any general distinguishing criterion could even exist. This is precisely where our contribution lies: our argument is that 1d stratification in order of energy content would be distinctly biotic. We revised the manuscript to emphasize this point (introduction lines 44-56, discussion lines 281-301).

The reviewer is also correct that this outcome requires resources to be supplied inhomogeneously in space, establishing the origin from which the spatial stratification would establish itself. We added an explicit statement to that effect (lines 319-321).

There are some minor suggestions for the readability of the paper.

(A) In Eq. (1), the competitive interaction between species is described by the matrix element α_{ij} . However, the resource is labeled by index α , and the use of the same symbol can cause confusion.

We fixed the notation, renaming species interactions to A. (revised Eq. 1)

(B) The resource flux is written as flux K but never used in the equations directly.

Yes, it is used in boundary conditions, which we should have included explicitly. Fixed! (new Eq. 3)

(C) The consumption of resource by each species is described by the matrix element $k_{i\alpha}$, which is never explicitly explained in words, and the readers are expected to deduce its meaning. It would be clearer to state its definition.

Yes, absolutely. We fixed this omission. (Lines 156-158)

(D) The authors talk about stratification, but I think it should be stated explicitly as “spatial stratification” to avoid any chance of a reader thinking about other stratifications related to internal variables such as metabolism.

Good point. We revised to include “spatial” throughout the manuscript. (e.g. lines 92, 190, 194, 209, 213, etc.)

(E) On lines 42-45 the authors state that “assumption of life forming an ecosystem is almost as weak as the minimal required assumption of a Darwinian process.” I had to read this several times before I understood that what they really meant was that it is generically the case that life forms ecosystems. When I first read it, I thought they meant that the assumption is weak, meaning that it does not have much explanatory power. I think they should just say that they assume that life forms ecosystems, since all examples on Earth have this characteristic, even in extreme environments such as acid-mine drainage.

Thank you for the feedback! We edited this sentence. (Lines 67-68)

(F) They give the example of an exception to the notion that ecosystems exhibit species richness, which is the work of Chivian et al (2008). However, even that supposedly single species ecosystem has organisms whose genome has clear evidence of horizontal gene transfer from Archaea to Bacteria. This indicates that the organisms detected had at some point in their past, if not in the present, been part of an ecosystem. Also, I wondered if one needs to distinguish between stable and unstable ecosystems. An ecosystem on the verge of extinction might be passing through a bottleneck where there is only a single species left.

This is great, thank you for helping us strengthen that argument! We revised lines 63-66.

Reviewer #2 (Remarks to the Author):

In this manuscript, the authors propose a new candidate for agnostic biosignature for the search for extraterrestrial life. The accumulation of specific materials has previously been proposed as a biosignature. However, these are based on the specific metabolic pathways of life found on Earth, and whether they can be applied to extraterrestrial organisms, whose metabolic pathways may be completely different from those on Earth, is unknown. Therefore, the authors proposed a new biosignature based on microbial ecosystems, the energy-ordered resource stratification. The authors constructed a minimal model of a microbial ecosystem and clarified the conditions under which spatial resource stratification emerges.

The authors' attempt to explore biosignatures beyond material details is very important. Also, the concept of biosignatures based on ecosystems is novel. I agree with the argument that the

distribution of compounds should be used as a biosignature rather than the presence of specific ones. However, the effectiveness of resource stratification as a biosignature is somewhat questionable. The authors need to address several issues.

Major:

1. First, to observe the resource stratification proposed by the authors as a biosignature, samples must be returned while maintaining a solid spatial structure, but is this even possible? The current method of sample return missions does not preserve the spatial structure. If the technology to return samples while preserving structure is established in the future, **it may be possible to observe extraterrestrial organisms directly [emphasis added]**. In that case, in what situations do the authors expect the proposed biosignature to be effective?

Thank you for prompting us to clarify this, as this is indeed a key point that our original manuscript failed to deliver clearly.

In the search for extraterrestrial life, the limitations are both technological and conceptual. Here, our contribution is to propose a new solution to the conceptual problem: what measurable signatures can be expected from a form of life *just* because it's life? In this version of the question, we have to ask: Would we, in fact, recognize life if we see it? What if this life operates at an unexpected scale? In this paper, our description is at least in principle compatible with, say, an "ecology" of atmospheric convection cells, each a mile in diameter, creating a layered structure in a planet's atmosphere...

Thus, in this work we use "measurable" as "measurable in principle", not "measurable remotely", and definitely not "measurable with current capabilities". Whether this idea can be adapted to a form compatible with the current technologies, and if so, in what context (ancient life on Earth, Solar system missions, remote sensing of exoplanets) is a question for the broad set of relevant experts. Our aim with this paper is to put this idea in front of these experts. We edited and expanded both the Introduction and Discussion sections to clarify our intent (introduction lines 30-38, 49-56, discussion lines 269-301, 350-365).

2. The authors assumed that the growth yield on a resource correlates with its chemical energy, but how true is this assumption? Is total chemical energy the internal energy of a compound or its chemical potential? If it is the former, then even if the internal energy is high and cannot be consumed by living organisms, it will not contribute to the growth rate. If it is the latter, then the chemical potential is determined by the amount of the compound present, but since this amount itself depends on the metabolism of the organisms. Is it possible to determine the chemical potential of a compound from a sample with unknown extraterrestrial organisms?

Thank you for prompting us to be more precise on this point. We edited to clarify that we meant the internal energy of the compound (lines 167-168, 301, 318). It is true that this may not always match the energy extractable by a living organism – and this caveat stands even without invoking the distinction between energy and chemical potential (e.g., it took evolution millions of years to learn to metabolize lignin, abundant during the Carboniferous period). Nevertheless, as we explain, overall it is plausible to expect the two quantities to be correlated, because an ability

to extract more of the available energy would be favored by selection (lines 178-179, 315-316). Thus, the Darwinian nature of the evolutionary process – the only thing we can postulate about life – works in our favor. Still, this is indeed an assumption; the revised manuscript emphasizes this point and highlights its centrality to our argument (lines 164-179, 315-316).

3. In their model, the authors assume the existence of competition that does not depend on competition for nutrients (depending on $\alpha_{\{i,j\}}$). They also show that if this competition is not strong enough, i.e., without global competition between organisms, resource stratification will not appear. Various competition mechanisms that work between specific species are known, but what kind of global competition mechanism is assumed?

Thank you. We clarified that we meant local competition such as contact-based inhibition or direct competition for physical space (lines 134-136), and added 3 references (New refs Coulthurst (2013), Hibbing et al (2010), Lloyd & Allen (2015)).

4. In the model, the authors assume that all species are specialists, i.e., they consume only one compound. If many species are generalists, how well will resource stratification be maintained? Good point, also raised by reviewer #1. We have expanded our model to allow for generalists, and show that stratification is robust for a range of the “extent of generalism parameter”, interpolating between pure specialists and complete generalists (line 267, new fig. S3, lines 596-610).

Minor:

Line 118: The authors stated, "If the system were well-mixed, the single “fittest” biological species using the most energy-dense resource would outcompete all others and take over the system." However, will this happen unless p is close to 1? For example, if p is 0, the same number of species as the number of nutrients would remain.

Thank you; this statement was indeed incorrect (and also unnecessary for our argument). We removed it (lines 180-182).

Reviewer #3 (Remarks to the Author):

This paper is part of a series of very interesting and important recent papers on microbial ecosystem self-organization, stability, energy extraction, and environmental perturbations by Goyal, Tikhonov, et al. This body of work introduces a variety of new general theoretical perspectives to the field, and is both useful for interpreting modern ecology and evolution and for understanding important questions about the origins of life.

However, the current paper is more of an initial hypothesis suited for a more specialized journal, and I don't think meets the level of Nature Communications.

Thank you for your time and input. Before describing how the specific comments below helped improve the manuscript, we would like to address this major point.

We did not select Nature Communications for its impact factor, but for the breadth of its audience. In fact, we believe it is the one and only journal where this paper could “work”.

There are two cases when a paper is suitable for a broad-audience journal. First, the achievement is important enough that everyone should hear about it. We make no such claim; our case is the second: the idea presented is simple enough and general enough that it could find use in a variety of contexts, provided it is seen by the relevant experts who can take it further.

In the search for extraterrestrial life, the limitations are both technological and conceptual. Our contribution is to propose a new solution to the conceptual problem: what measurable signatures can be expected from a form of life *just* because it's life? Even if “measurable” is taken as “measurable in principle”, not “measurable remotely” or “measurable with current technological capabilities”, this question is very difficult – a priori, not obviously solvable! – and this is the version of the question we consider.

Our work is not an initial hypothesis: it presents precisely one idea (self-replication + ecology leads to energy-ordered spatial stratification), but this idea is fully explained and supported, and having read all the specific references mentioned by the referee & other work by those authors (which was immensely helpful), we believe our contribution still stands (see below). In our minds, the fact that the idea can be fully explained and supported by a very short argument (abiotically, reaction rates and yields are uncorrelated; biotically, standard models all generically give rise to the signature, and for simple enough reasons that the generality is clear) makes our paper more appropriate for a broad journal, rather than less.

Now, whether this idea can be adapted to a form compatible with the current technologies, and if so, in what context (ancient life on Earth, Solar system missions, remote sensing of exoplanets) – these questions are for the experts of those respective fields, and any claims coming from us would indeed be pure hypotheses. In summary, we agree with the reviewer in everything but their conclusion: In our minds, their argument is precisely why Nature Communications is the uniquely appropriate venue. The journal self-defines its niche with three bullet points (copied verbatim from <https://www.nature.com/ncomms/journal-information>; emphasis added):

- Publishes in all areas of life, health, social, physical, chemical and Earth sciences
- Presents **important advances of significance to specialists** within each field
- High visibility with open access

Note the absence of superlatives (e.g. “highest-impact”), making this venue uniquely suited for disseminating an interesting idea that would resonate with a broad audience of relevant experts who could take it further.

We revised the intro and discussion to clarify our scope and intent (see the PDF highlighting our extensive edits), and incorporated the suggestions and references below, which helped contextualize our presentation. We believe the text does not oversell the conclusions, is upfront about limitations, and we would be happy to let the reader judge the value of our contribution.

I have two major comments that should be addressed in any case:

1) The work needs to be much better situated in the astrobiological literature:

- A classic focus in astrobiology-related experimentation and field work has been metabolic stratification in biofilms by Tori Hoehler et al. and many other types of ecosystems including ocean sediments (Bo Jorgensen, Jan Amend, Victoria Orphan, etc.). Funding studies on low-energy life, the diversity of energy harvesting mechanisms, and energetic ecologies is a long-running focus of NASA and there is much more work to cite in this study.

We thank the reviewer; in scouring the literature, we did our best, but evidently not enough, and these additional references were very helpful. We went through the publications by these researchers and agree this work is highly relevant. Incorporating these citations helped us better focus and contextualize the revised manuscript. To the best of our understanding, our contribution still stands. The closest statement in the prior literature is the following passage in the paper “A ‘Follow the energy’ approach” by Hoehler et al. (emphasis added): “*Where the activities of energy harvesting or energy investment are (or can be) demonstrably distinct from equivalent abiotic processes of energy flow, biosignatures potentially exist. The detectable products of such energy investment are thermodynamic disequilibrium [...], high information content [...], and **physically or chemically ordered systems or structures.***” This quote is indeed extremely relevant; however, no criterion distinguishing biotic and abiotically generated order was proposed. Hoehler et al explain what kind of a biosignature may exist, “if the two kinds of order were found to be demonstrably distinct.” We here propose a specific instantiation, and link its emergence to ecology. While metabolic stratification in ecosystems is indeed much studied, we were unable to find any paper proposing that this stratification (without reference to a specific metabolism) could serve as a biosignature. From all our reading, the Hoehler et al. quote above came closest, but was never expanded upon.

The revised manuscript uses these citations (13 new references: new refs. 10, 28-29, 31-35) to better contextualize our contribution (introduction: lines 39-58, 85-92; discussion: lines 269-301; 324-336; and 350-365). Clarifying these links to existing literature certainly helped focus the manuscript; but we believe it still adds value.

- There are many recent efforts focused on energy gradients as an agnostic biosignature (see the work of Peter Girguis and Jeffrey Marlow) which is the main proposal of this paper.

Thank you; clarifying this point was very important. While the term “energy gradient” is highly prevalent in the astrobiological literature, (in our understanding) it refers to the fact that sustaining life requires the presence of a compound at a chemical disequilibrium, which could be harnessed for energy. This “gradient” is in energy space. In contrast, we are talking about a **spatial** gradient of multiple energy-containing compounds. We revised the text to clarify this contrast, adding a footnote (included into the reference list as Ref 19 in the PDF highlighting changes) and replacing “stratification” by “spatial stratification” (e.g. lines 92, 190, 194, and elsewhere throughout the manuscript). We also included references to work by Girguis and Marlow (Marlow et al 2018, Marlow et al PNAS 2021, Marlow et al Env Microbiol 2021).

- Ecological perspectives on astrobiology have recently had a few important advances both conceptually and specifically that should also be interfaced with in this study. Recent prominent examples include:

a) Affholder, A., Guyot, F., Sauterey, B., Ferrière, R., & Mazevet, S. (2021). Bayesian analysis of Enceladus's plume data to assess methanogenesis. *Nature Astronomy*, 5(8), 805-814.

b) Kempes, C. P., Follows, M. J., Smith, H., Graham, H., House, C. H., & Levin, S. A. (2021). Generalized stoichiometry and biogeochemistry for astrobiological applications. *Bulletin of Mathematical Biology*, 83(7), 73.

c) Meurer, J. C., Haqq-Misra, J., & de Souza Mendonça, M. (2024). Astroecology: bridging the gap between ecology and astrobiology. *International Journal of Astrobiology*, 23, e3.

d) Cockell, C. S., Simons, M., Castillo-Rogez, J., Higgins, P. M., Kaltenegger, L., Keane, J. T., ... & Vance, S. D. (2024). Sustained and comparative habitability beyond Earth. *Nature Astronomy*, 8(1), 30-38.

Thank you; we have incorporated them in the text (new refs 13-17).

2) It is not clear to me how these models are differentiated from any number of ecological models, even those used in modeling detailed global ecology. A broader discussion and explanation is needed here.

Thank you for prompting us to clarify. We did not seek to build a different ecological model; rather, our intent was to show that the emergence of energy-ordered resource stratification is generic, and naturally emerges even in the simplest models commonly used in the field. This has been clarified (lines 255-267, 302-323), and we have added 3 model generalizations (redundancy, generalism, and cross-feeding, all known to be ubiquitous and commonly modeled) to illustrate this further (255-267, new SI sections lines 544-610, new SI figures S1, S2, S3).

Reviewer #4 (Remarks to the Author):
